# What Can Synergetics Contribute to Embodied Aesthetics?

**DOI:** 10.3390/bs7030061

**Published:** 2017-09-02

**Authors:** Hermann Haken

**Affiliations:** Institute for Theoretical Physics, Center of Synergetics, Pfaffenwaldring 57/4, Stuttgart University, D-70550 Stuttgart, Germany; cos@itp1.uni-stuttgart.de or haken@theo1.physik.uni-stuttgart.de; Tel.: +49-(711)-685-64990; Fax: +49-(711)-685-64909

**Keywords:** pattern recognition, order parameters, aesthetics

## Abstract

I deal with perception and action (e.g., movements) using results from synergetics, a comprehensive mathematical theory of the self-organized formation (emergence) of spatial, temporal, or functional structures in complex systems. I illustrate basic concepts such as order parameters (OPs), enslavement, complexity reduction, circular causality first by examples of well-known collective, spontaneous modes of human behavior such as rhythmic clapping of hands, and so forth, and then by face recognition. The role played by OPs depends on context. In the case of face (or pattern) recognition an OP represents the concept of an individual face (action of mind) and it enslaves the action (firing rates) of neurons (body). This insight allows me to interpret syndromes as order parameters playing their mind/body double role. I present criteria for the identification of OPs and discuss their general properties including error correction and remedy of deficiencies. Contact is made with a recent paper by Sabine Koch on embodied aesthetics. My approach includes the saturation of attention at various time scales (ambiguous figures/fashion). Adopting a psychological perspective, I discuss some ingredients of beauty such as proportionality and symmetry, but also the importance of irregularities.

## 1. Introduction

While the readers of my paper are surely familiar with what behavioral science is about, this need not be so concerning synergetics and why it may be of interest to behavioral science.

Synergetics (“science of cooperation”) [1,2] is an interdisciplinary field of research that deals with complex systems that can spontaneously produce spatial, temporal, or functional structures (Figure 1). The considered systems are composed of many individual parts that exchange energy, matter, and/or information among each other, as well as with the surrounding of the total system (Figure 2). 

These systems acquire their structures by self-organization, that is, without any specific interference from the outside, say, by a sculptor. Synergetics is grounded in mathematics (dynamic systems theory, theory of stochastic processes, phase transition theory, and information theory) and has developed its own specific concepts. Synergetics has found numerous applications in physics, chemistry, biology and, in particular, behavioral science, for example, movement coordination and recognition of faces, facial expression and of movements. Thus, a common ground between behavioral science and synergetics is—I think—the study of the relations between a specific “behavioral pattern” as a *whole* and its *individual* constituents. The *whole* may be a social *group* and the *individual* constituents persons, or the *whole* may be a human *brain* and the constituents its *neurons*, just to mention two examples that are relevant for my paper. The central aim of synergetics is to cast these relations, based on realistic models, into a mathematical form which allows quantitative treatments. As it has turned out there is a clear-cut correspondence between a conceptual approach that can be verbalized, and a formal mathematical treatment. In my paper I will present the conceptual approach. My paper is organized as follows. Section 2 has little to do with synergetics. Nevertheless, I state that (embodied) aesthetics necessitates a definition of beauty, but argue why there are no criteria for beauty. Section 3 presents basic concepts of synergetics, starting from well-known examples of collective human behavior, and then dealing with face recognition and movement coordination.

Section 4 deals with order parameters and embodiment, while Section 5 returns to the question of beauty and its psychological consequences. The concluding Section 6 makes contact with a recent paper by Sabine Koch [3] on embodied aesthetics. Her paper also contains numerous references that may serve as background information for parts of my paper that deal with beauty and/or embodiment. I assume that my readers are familiar with her paper so that I do not duplicate these references.

## 2. What Is Beauty?

Nature abounds with objects that we perceive as beautiful, such as flowers, animals, landscapes. We look at beautiful artefacts such as paintings, sculptures, architecture and we may be delighted by poetry and music. We may perceive movements as beautiful, such as the elegant trot of a horse, and still more the dancing of people. At a more abstract level, mathematicians speak of beautiful formulae, and physicists may consider their laws as beautiful. And more generally, people are delighted by new insights, or discoveries. Yet, a scientific study of aesthetics and its embodiment requires a definition of beauty. Whenever I met the famous art historian Sir Ernest Gombrich (1909–2001) who had studied psychology, I asked him: What are the criteria for beauty? Each time he replied: There are no criteria for beauty! Why? Surely there are several reasons for that. Here I want to discuss one reason that appears to me most likely. To this end, let us forget “beauty” for the moment and consider an ambivalent picture such as Figure 3. 

Our perception oscillates back and forth between the interpretations “vase” or “faces”. In 1920, the Gestalt psychologist Wolfgang Köhler [4] proposed an explanation: When we have recognized an object, our attention for this object fades away and we are able to recognize the other object, and so on (a detailed mathematical model of this process was presented by Ditzinger and Haken [5]). While here the switching time between percepts is a few seconds, in other cases it may be much longer. A typical example is provided by fashion. First a new style is created, which is followed by an ever increasing number of people—until they get bored and require a new style. Thus an excitement/boredom cycle results. This deep-rooted human trait is well reflected by the Latin proverb “variatio delectate” or by poetry, for example, by Goethe’s Faust: “*Werd ich zum Augenblicke sagen: Verweile doch! du bist so schön! Dann magst du mich in Fesseln schlagen, dann will ich gern zugrunde gehen!/When I say to the moment flying: Linger a while—thou art so fair! Then bind me in thy bonds undying, and my final ruin I will bear!*”

In Section 5, I will discuss what that means for some ingredients of beauty. However, more importantly, we must be aware of at least one conclusion for the relation between therapist and client: Does their understanding of what beauty is coincide? And, does it persist (e.g., does boredom arise?) In the present context we may interpret “beauty” in a wide sense as the property of an object or happening that causes good feelings, excitement, delight, harmony, etc. in the recipient.

## 3. Basic Concepts of Synergetics

Synergetics deals with complex systems, that form spatial, temporal, spatio-temporal or functional structures by self-organization. In the present context, a number of insights gained by the mathematical approach of synergetics can be visualized by examples of spontaneous collective human behavior.

(a) *Formation of spatial structures (patterns)*

Consider swimmers in a swimming pool. If there are few swimmers, they may swim quite independently of each other in various directions (disordered state). If the number of swimmers, that is, their density increases beyond a critical value, in the disordered state the swimmers impede each other so much that they acquire an ordered mode, namely, they swim in circles. This may happen spontaneously, that is, without any given orders from the outside. First, few swimmers begin swimming in circles, then more and more swimmers join them (Figure 4). 

Anyone who enters the pool is automatically forced to join the collective, ordered motion. Spatial patterns are well known in the animal kingdom: flocks of birds, schools of fish, to mention but two examples.

(b) *Formation of temporal structures (patterns)*

When, after a concert, the listeners get enthusiastic, quite often the following happens: While first the clapping of hands is quite irregular, suddenly few people start clapping their hands rhythmically until very quickly the whole audience claps rhythmically. A further example is provided by people walking across an elastic bridge that can oscillate. Then quite often they synchronize their pace so to enhance the oscillation of the bridge. They enjoy feeling the resonance! An example of the animal kingdom is provided by the rhythmic blinking of fire flies.

(c) *Formation of spatio-temporal structures (patterns)*

Spectators in an arena may spontaneously form a “la ola” wave. First, along some line, people stand up, others at their side follow and then those who got up first, sit down. In this way people generate a “wave” running around the arena. These simple examples may allow me to formulate general concepts of synergetics.
(1)Control parameters

They are fixed quantities that are externally or internally prescribed. In the example (a) the control parameter is the density of swimmers; in the examples (b), (c), it is the internal excitement of the participants. When a control parameter exceeds some threshold, the old state of the total system is abandoned and replaced by a new one. In other words,
(2)the old state becomes *unstable*.(3)Close to such instability “points” the formation of a new structure (pattern) is triggered by some spontaneous events (e.g., rhythmic clapping of hands of only few people).(4)The new ordered state is characterized by an *order parameter*. It plays a double role: it describes the macroscopic state and it prescribes the action of the individual parts.(5)*Slaving principles*.

One or few order parameters determine (enslave) the behavior of the individual parts (Figure 5) (e.g., swimmers follow the circular motion, listeners join the collective hand clapping).

These examples may indicate that there are different grades of the “strength” of “enslavement”. I have discussed the sociological implications of the “enslavement” notion at various occasions, because many sociologists strongly criticize this notion stressing that humans are free in their decisions.

(6)Complexity reduction

Instead of describing the actions of the numerous parts it suffices to deal with the order parameter(s). Complexity reduction is crucial to face (or pattern) recognition, movement control, and sensory/motor action, among others.

(7)Circular causality

This concept is rather counterintuitive because it contradicts our understanding of the conventional cause—effect relation. However, in the case of self-organization the order parameters enslave the individual parts, while the parts through their joint action, generate the order parameters. However, just think of the hand clapping: each participant *hears* (perceives) the collective hand clapping, but at the same time she/he *produces* this effect.

### Visualization of the “Behavior” of Order Parameters

In quite a number of cases the “behavior”, or precisely speaking, the dynamics of order parameters can be visualized by an *attractor landscape*. An example is provided by Figure 6. 

On the abscissa, we plot the size of the considered order parameter, OP. We represent its motion (dynamics) by the motion of a ball that rolls down a grassy hill so that it may come to rest at the bottom of the valley. The ordinate, called V, may be identified with the height of the position of the ball. 

Figure 6.1 represents a situation where the OP is zero, indicating a structureless or disordered state (think of the irregular motion of swimmers in a pool). When a control parameter is increased (e.g., density of swimmers), in the mathematical model the bottom of the valley becomes quite flat. (Figure 6.2). Since the ball is subject to random fluctuations (“kicks”), it moves randomly considerably away from the position of OP = 0. This effect is called *critical fluctuations* and is a precursor to *phase-transitions* where a new structure appears (Figure 6.3). That is to say that with further increase of the control parameter two new valleys appear that entail two new stable values of the OP, of which only one can be realized. In the concrete “swimmer model” this means that the swimmers now swim clockwise *or* anticlockwise. The observation of critical fluctuations of the behavior of patients may play an important role in neurology (e.g., in a number of cases such fluctuations may be indicators of evolving epileptic seizures). The attractor landscape model may serve to discuss “saturation” effects. According to Tschacher [6], the slope of a valley may represent the degree of intentionality acting as control parameter (Figure 7a). 

When the corresponding task has been fulfilled and the control parameter is zero, the slope vanishes (Figure 7b). Attractors that vanish while the corresponding OP is built up have been termed quasi-attractors by Haken [7], in particular in the context of the recognition of ambiguous patterns. First, in the presence of the corresponding valley, an OP increases beyond a critical size (the conscious state) until the valley and with it the OP vanish. After some time the valley is reestablished.

## 4. Order Parameters and Embodiment

To elucidate the role of order parameters in embodiment, we consider the example of face recognition. In this case, an observer is able to recognize a face though it may be partly obscured. He/she is able to complete this percept in his/her mind and associate a name with this face. Thus face recognition is conceived as action of associate memory, or, on a more abstract level, as the completion of a set of incomplete features (data) to a set of complete features (data) according to stored prototype patterns [8]. 

An example is shown in Figure 8, where the family names are encoded by letters. The individual patterns are represented by their pixel grey values. The grey values of any shown image (Figure 9, left hand side) are encoded as neural excitations that are transmitted to the neural network of the brain (Figure 9, right hand side), where they are further processed.

The pattern recognition process is modeled by a pulse-coupled neural network that I have dealt with elsewhere in great detail [9]. In the context of my present paper, the following results are relevant: As it turns out, consistently with the mathematical treatment of the network, we may ascribe an order parameter *ξ_k_* to each pattern *k*=1,···,5 (Figure 8). When an incomplete pattern (Figure 10) is offered to the network, a competition between the order parameters evolves that can be visualized as motion of a representative point (ball) in an attractor landscape. 

This competition is eventually won by an order parameter that “enslaves” the individual neurons of the network so that a complete pattern emerges (Figure 11). 

To put it cautiously: these results strongly suggest that the order parameters play a double role—they represent the perceived object, that is, action of *mind*. Simultaneously they enslave the collective action of neurons: a *body-action* (Figure 12). 

My approach strongly supports Spinoza’s view that mind and body (matter) represent the two sides of the same coin (which I identify with an order parameter). Quite clearly, this view is in contrast to Descartes’s view on the dichotomy between mind and body. So far, I have represented the example of *visual* patterns. My approach holds also for movement or behavioral patterns. In all these cases, the prototype patterns may be learned or can be of an archetypical nature.

Since the order parameter concept holds also for group dynamics (of people or animals), it might be an interesting *speculation* to ascribe some kind of mind also to groups, some kind of consciousness to collectives, so to speak. While in psychology we can verify the concept of consciousness by personal individual introspection, this “tool” is absent in sociology. It is important to note that the role (meaning) of order parameters (OPs) depends on the *context*. To underline this statement, consider *movement* patterns produced by a human with his/her coordination between limbs and/or body motion. An explicit example is provided by experiments of Kelso [10], who instructed subjects to move their index fingers in parallel at a slow speed (Figure 13). 

When the prescribed speed was increased more and more, quite involuntarily the parallel movement switched to an antiparallel, symmetric one. This spontaneous switch was modeled by means of an order parameter dynamics in an approach by Haken, Kelso, Bunz [11] that has become paradigmatic in the field of movement coordination. In this model, the order parameter can be identified as relative phase between the index fingers. In further experiments by Kelso and his coworkers and the theoretical work of my group, we could demonstrate that acoustic perception, movement coordination, and activity of the CNS (central nervous system) are governed by only two order parameters [10]. 

These OPs govern at the next level perception and action, each represented by the corresponding OPs, which in turn “enslave” the elements participating in perception and the elements participating in action (Figure 14).

## 5. Criteria for OPs and Syndromes

For a conceptualization and mathematical modeling we need criteria for OPs. Quite generally, OPs refer to the total system. In particular, they refer to the coordination (spatial, temporal, functional) between the parts of the studied system. Moreover, a time-scale separation holds. That is to say, OPs change slowly, whereas parts change fast. This can be checked by means of their reaction time to perturbations. Additionally, while a change of OPs causes a change of the behavior of the parts, a change of a part (or a minority of parts) has practically no effect on the OPs (just think of the hand clapping example!). The joint action of an order parameter and the slaving principle is visualized in Figure 15. 

In particular, we may ascribe measurable values *v_l_,l*=1,···,*N* to the individual parts. An order parameter governs a whole configuration (“pattern”) of *vs*. If a configuration is incomplete, the order parameter serves for a complete configuration or it is able to correct errors (e.g., of texts). It can serve for synchrony between the parts in oscillating systems. Some time ago I suggested that an order parameter represents a syndrome, where the *vs* represent a set of manifestations that may belong to mind or body (Figure 16).

Again, a syndrome may serve for completion of manifestations and synchrony. Order parameters may even “govern” interpersonal relations, for example, in an alliance between client and therapist. A central question concerns the quantification of emotions and their measurements. Well-known examples are measurements (observations) of skin resistance, heartbeat, blood pressure, “goose skin”, eye movements, but also analysis of facial expressions (e.g., Ekman classification) or/and of gestures, and speech characteristics. EEG, MEG, fMRI measurements allow the study of brain activities, especially of their localization and correlations. By these and other methods (e.g., extra-cranial excitations) specific areas could be identified, for example, nucleus accumbens (enjoyment), amygdala (disgust, fear). The data can be supplemented by personal ratings. While I think it is at least plausible to consider *intrapersonal* ratings as (more or less) reliable, I believe that *interpersonal* ratings are highly questionable due to the qualia problem. Based on quantification, promising models of psychological processes have been developed by Schiepek et al. [12], Tschacher et al. [13] and others.

## 6. Some Basic Ingredients of Beauty

Though I have stressed the absence of criteria for beauty (Section 2) there seem to be at least some ingredients of beauty that have endured over the centuries. One aspect is *proportionality*, the other one *symmetry*. 

In the latter case, there may be mirror symmetry (Figure 17), translation symmetry (Figure 18) or rotation symmetry (Figure 19). 

The latter symmetries are connected by repetitions. Interestingly, these symmetric patterns can be observed in many self-organizing systems ranging from fluids over macroscopic chemical reactions to plants and animals. What is the psychological impact of such regularities? They strengthen confidence, they make us calm. This is in line with the idea that the human mind, consciously or unconsciously, is all the time trying to unearth (general) rules that allow it to extrapolate, that is, to foresee, what will be next. Obviously, regularities facilitate this process. Incidentally, however, (too much) regularity generates boredom! Thus, an artwork needs “surprises”, namely, breaking of symmetry (as exemplified by Picasso’s faces), irregularities, unusual combinations of elements, and so forth. The weights of order and disorder, expectation and surprise are changing in the course of time, though each epoch seems to prefer a specific balance between order/disorder. Clearly, my remarks have some consequences on therapies based on embodied aesthetics—for some clients it may be preferable to strengthen the “order/regularity”, for others, the "excitement/variation” aspect.

## 7. Conclusions

In my paper I have tried to give a brief outline of the concepts and results of synergetics. According to it, external conditions (represented by control parameters) may lead to a qualitative change of the macroscopic behavior of a complex system. At least at these instances the behavior of the system is governed by one or very few order parameters that govern (“enslave”) the individual parts via circular causality. In this way, a perception-action cycle is governed by an order parameter. This implies that this cycle can be initiated by the impact of a control parameter on *action* or *perception*, or both of them. This order parameter approach can be immediately transferred to the circular model of embodied aesthetics by Sabine Koch [3], where the interplay between person and art (mediated by impression and expression) is now (in my approach) governed by an order parameter. As it seems to me, in this way the “re-invokation of the old concept of the ‘soul’ in order to grasp the holistic aspects of the arts”, (as suggested by Sabine Koch) may be substantiated. In the process, the therapist acts as “control parameter” by his/her incentives, providing specific surrounding, and so forth. In conclusion, a warning should be added: The order parameter concept is seductive because of its seeming generality. So always some care must be exercised when applying it.

## Figures and Tables

**Figure 1 behavsci-07-00061-f001:**
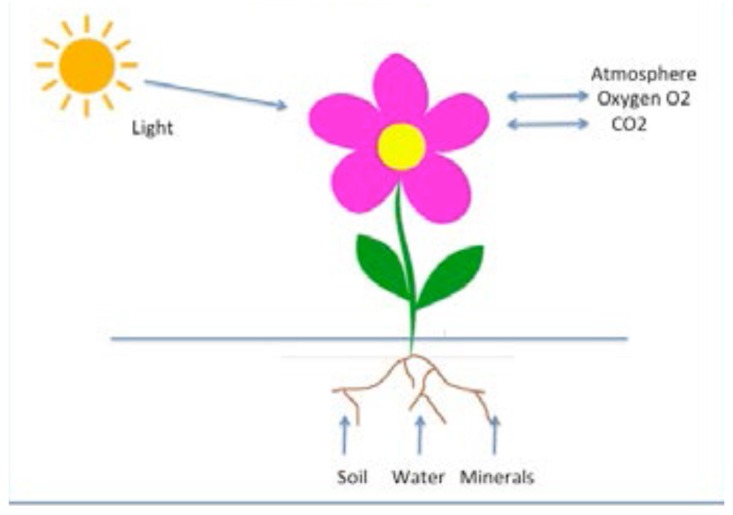
A flower is a typical example that exchanges energy and matter with its surround. Information is stored in its genes.

**Figure 2 behavsci-07-00061-f002:**
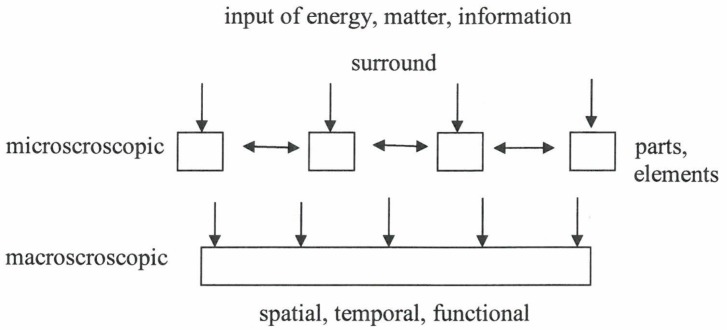
Scheme of a system that forms structures.

**Figure 3 behavsci-07-00061-f003:**
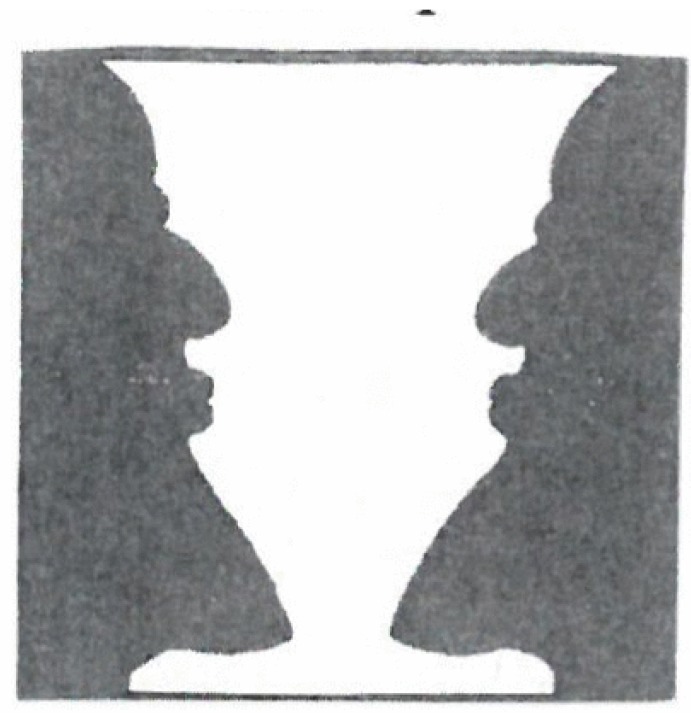
Vase or faces?

**Figure 4 behavsci-07-00061-f004:**
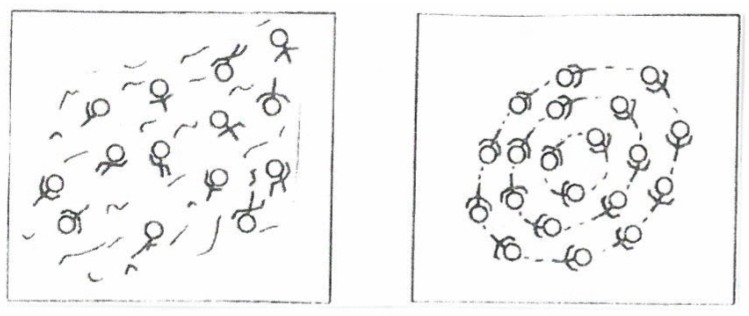
Transition from disorder to order: swimmers in a pool.

**Figure 5 behavsci-07-00061-f005:**
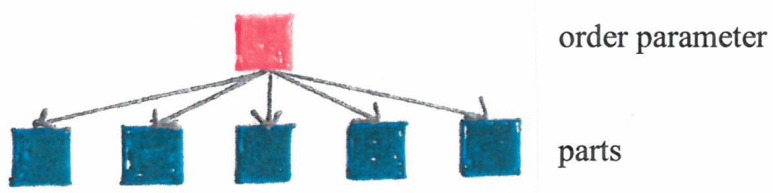
Slaving principle: One or few order parameters determine behavior of parts.

**Figure 6 behavsci-07-00061-f006:**
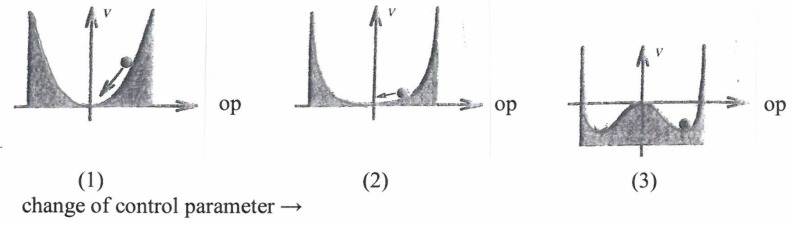
Change of order parameter landscape with increasing control parameter.

**Figure 7 behavsci-07-00061-f007:**
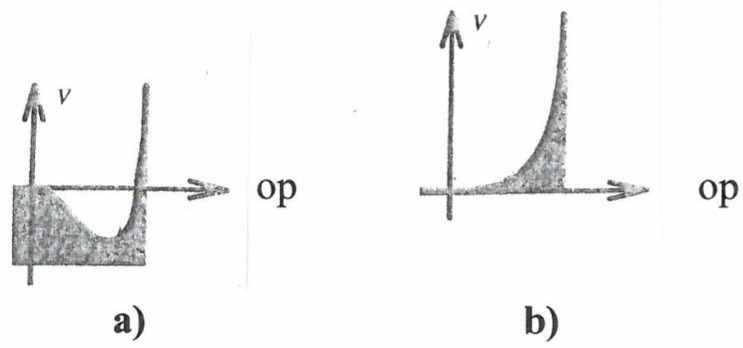
Vanishing of attractor. (**a**) Nonvanishing slope indicating degree of intentionality; (**b**) Slope vanishes after completion of task.

**Figure 8 behavsci-07-00061-f008:**
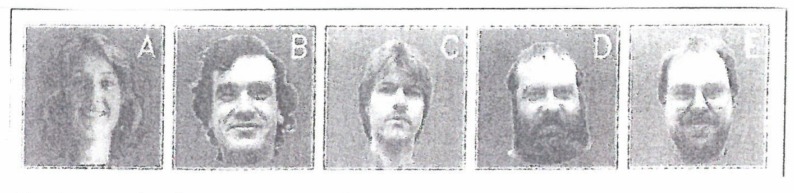
Example of stored (or learned) prototype patterns.

**Figure 9 behavsci-07-00061-f009:**
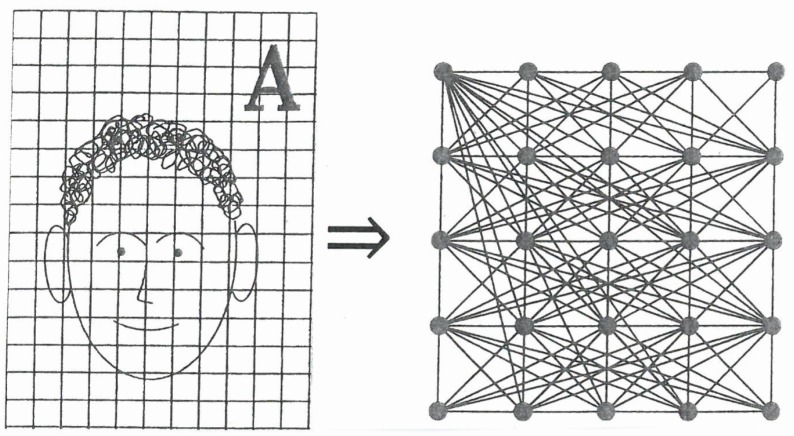
On the left-hand side, a scheme of an image with pixels. In the right upper corner, the family name is encoded by a letter. On the right-hand side, a schematic drawing of a neural network.

**Figure 10 behavsci-07-00061-f010:**
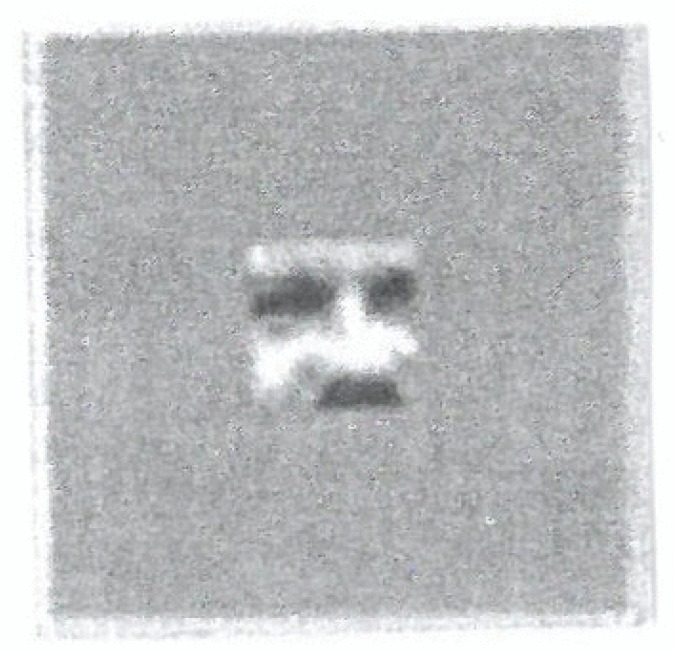
An incomplete pattern.

**Figure 11 behavsci-07-00061-f011:**
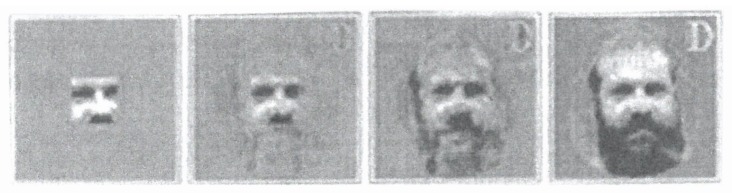
Completion of pattern by synergetic computer (modelling brain activity).

**Figure 12 behavsci-07-00061-f012:**
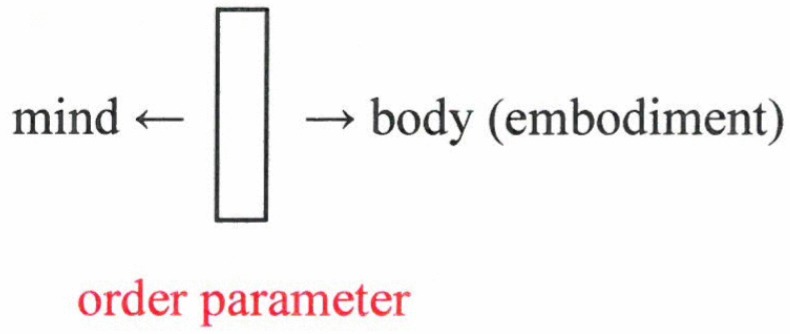
The two sides of a coin (the order parameter).

**Figure 13 behavsci-07-00061-f013:**
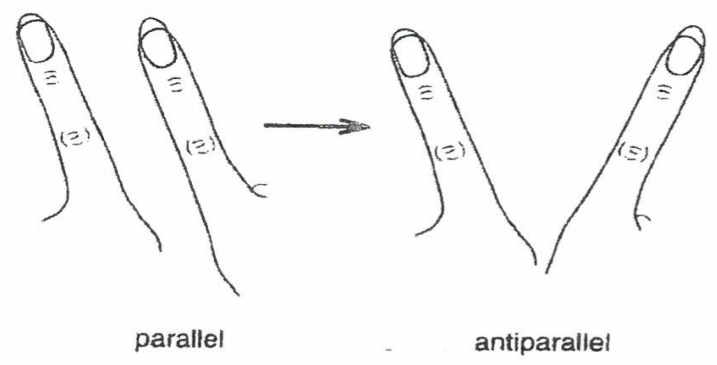
Kelso’s experiment: Spontaneous change from parallel to antiparallel (symmetric) movement of index fingers with increased speed.

**Figure 14 behavsci-07-00061-f014:**
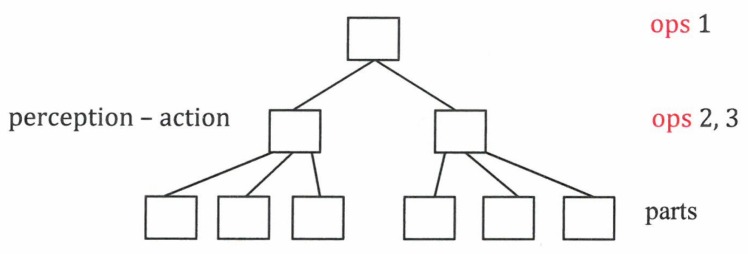
Order parameter hierarchy.

**Figure 15 behavsci-07-00061-f015:**
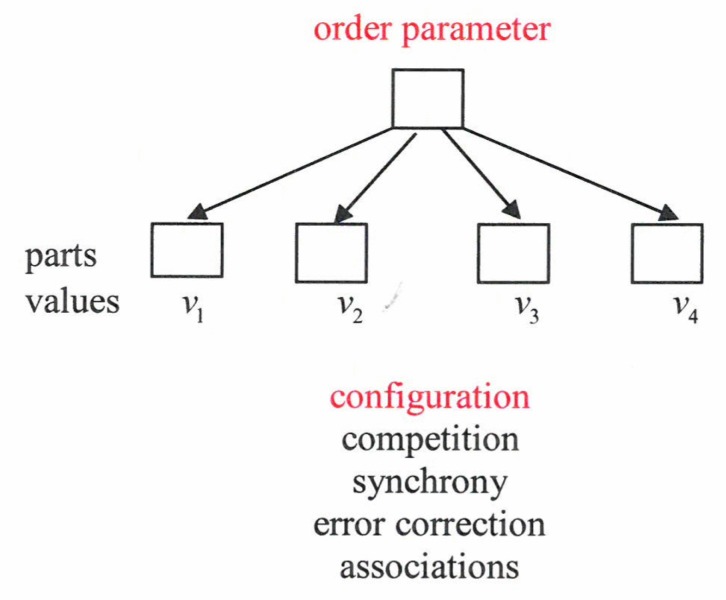
An order parameter governs a whole configuration of individual parts.

**Figure 16 behavsci-07-00061-f016:**
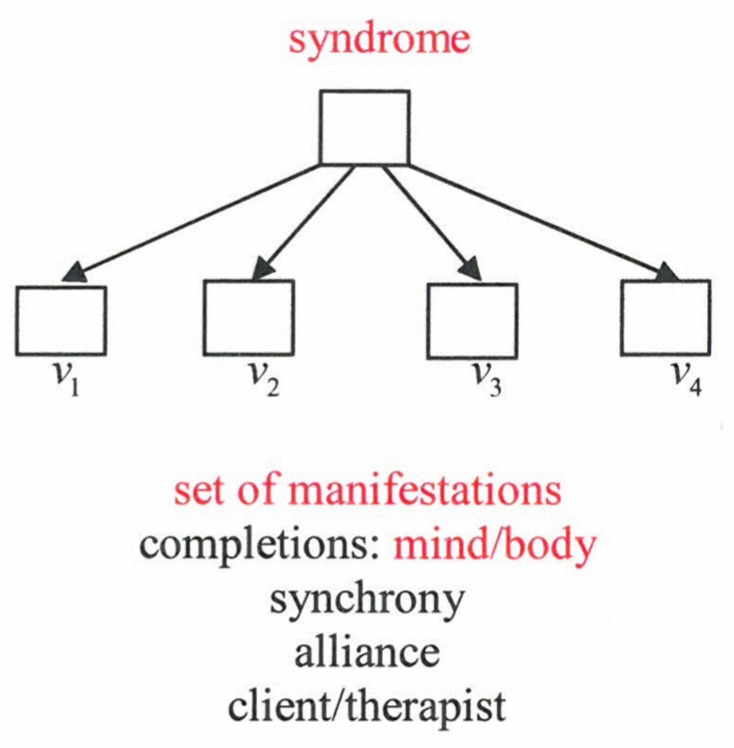
A syndrome plays the same role as an order parameter.

**Figure 17 behavsci-07-00061-f017:**
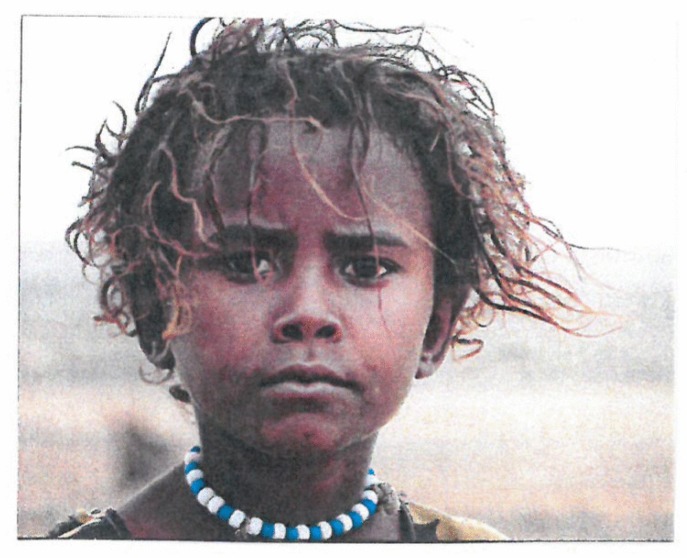
The human face is highly mirror symmetric.

**Figure 18 behavsci-07-00061-f018:**
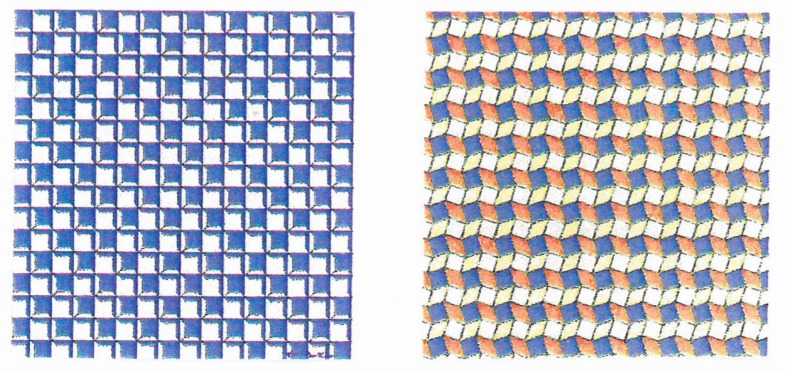
Examples of patterns produced by the translation of a square.

**Figure 19 behavsci-07-00061-f019:**
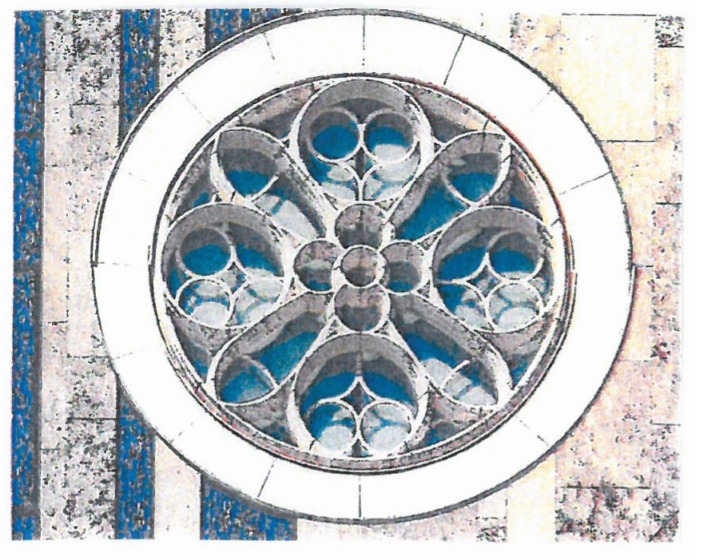
A rosetta as an example of rotation symmetry.

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
