# Peer review of "What Can Synergetics Contribute to Embodied Aesthetics?"

_behavsci, 2017, doi:10.3390/bs7030061_

Round 1

Reviewer 1 Report

This paper summarizes in verbal form what is actually a detailed mathematical and physical analysis of complex systems by the author that encompasses several disciplines, including, physics, neurobiology, psychology, cognitive and computer science. Although much of this work is known and has been published before, a novel aspect discussed here concerns the topic of aesthetics and beauty, and more conceptually the issue of the relationship between mind and body. The paper reaches out to a new audience in a very accessible way, characteristic of the masterly grasp of its content by the author. There are numerous minor errors that should be fixed prior to publication. See below:

Minor points:

At least in English self-organization is usually hyphenated and not a single word. Similarly, for Gestalt psychologist (l.81).

Spell out a.s.o.(l.82)

Quotation marks used throughout should be as follows: “xxx”, not ,,xx” (e.g. l.88)

l.108 in various directions. (disordered state). Remove first period.

l.124 rhythmically (sp.) not rhythmitically. Regarding the section (b) Formation of temporal structures, it would be nice to cite some references.

l.138 excitement (sp.)

l.155 criticize (sp.)

l.196 With respect to epileptic seizures, perhaps neurology is better than psychotherapy.

1.299 The word Syndrom. Does the author mean syndrome? If so, change spelling throughout.

l.342 “ranging from fluids to macroscopic chemical reactions…”

l.356 “According to Synergetics…

l.365 invocation (sp.)

l.368 conclusion (sp.)

l.368 The order parameter concept is seductive because…

Author Response

Thank you very much for your helpful review. I have checked language and performed the necessary corrections in the revised manuscript 

Reviewer 2 Report

I very much enjoyed reading this article. It is inspiring to see that the concepts of synergetics can even be applied to aesthetics and the perception of beauty. The article brings together nicely the theory of synergetics in a nutshell with  behavioral sciences and aeasthetics.

minor misprints: 

line 305: completition

line 368: The order parameter concept is a seducive because

I recommend the acceptance for publication

Author Response

Thank you for you review. I have corrected errors in the revised manuscript (R1)